# Psychometric Properties of the Persian Pittsburgh Sleep Quality Index for Adolescents

**DOI:** 10.3390/ijerph17197095

**Published:** 2020-09-28

**Authors:** Azita Chehri, Serge Brand, Nastaran Goldaste, Sodabeh Eskandari, Annette Brühl, Dena Sadeghi Bahmani, Habibolah Khazaie

**Affiliations:** 1Department of Psychology, Kermanshah Branch, Islamic Azad University, Kermanshah 6714673159, Iran; azitachehri@yahoo.com (A.C.); nastarangoldasteh@gmail.com (N.G.); 2Sleep Disorders Research Center, Kermanshah University of Medical Sciences, Kermanshah 6719851151, Iran; sodabeh.eskandari@gmail.com (S.E.); dena.sadeghibahmani@upk.ch (D.S.B.); hakhazaie@gmail.com (H.K.); 3Health Institute, Substance Abuse Prevention Research Center, Kermanshah University of Medical Sciences, Kermanshah 6719851151, Iran; 4Stress and Sleep Disorders (ZASS), Center for Affective, University of Basel, Psychiatric Clinics (UPK), 4002 Basel, Switzerland; annette.bruehl@upk.ch; 5Division of Sport Science and Psychosocial Health, Department of Sport, Exercise and Health, University of Basel, 4052 Basel, Switzerland; 6School of Medicine, Tehran University of Medical Sciences, Tehran 1416753955, Iran; 7Exercise Neuroscience Research Laboratory, The University of Alabama at Birmingham (UAB), Birmingham, AL 35209, USA

**Keywords:** adolescents, sleep patterns, Pittsburgh Sleep Quality Index, psychometry, Persian version

## Abstract

Background: Both cross-sectional and longitudinal studies show that poor sleep is a health concern related to further psychological and physiological issues during adolescence. To assess subjective sleep quality and sleep patterns among adults, the Pittsburgh Sleep Quality Index (PSQI) is a well and internationally established tool. Here, we established the psychometric properties of the Persian version of the PSQI for adolescents. Method: A total of 1477 adolescents (mean age: 15.47 years; 53.2% females) took part in the study. They completed a booklet on sociodemographic information, the Persian version of the PSQI for adolescents, and the Adolescent Sleep Hygiene Scale (ASHS). We relied on classical test reliability approaches of exploratory and confirmatory factor analyses. Results: Classical exploratory factor analysis yielded the seven-factor solution, with concurrent confirmation and overlap with the dimensions of the ASHS, although correlation coefficients were small to medium. A further factor analysis yielded a four-factor solution, explaining 72% of the variance of the PSQI. Further, three out of these four factors predicted the ASHS overall score. Conclusions: The Persian version of the PSQI for adolescents showed satisfactory psychometric properties. It follows that the Persian PSQI is a suitable tool to assess sleep quality and sleep patterns among adolescents.

## 1. Introduction

Restoring sleep has a fundamental influence on favorable brain functioning, and this holds particularly true for the developing adolescent brain [1,2,3]. Not surprisingly, also among adolescents, poor sleep and poor behavior are associated. To name a few examples, low sleep quality was associated with higher drug use (tobacco, cannabis, alcohol) both cross-sectionally [4,5] and longitudinally [6], higher risk-taking behavior [7,8], suicidal behavior [9,10,11], poor emotion processing [12], and increased daytime sleepiness [13]. Thus, there is sufficient evidence that among adolescents, low sleep quality is associated with unfavorable psychological functioning and behavior. Importantly, research showed that sleep duration and cognitive–emotional functioning are associated in a non-linear fashion. To illustrate, Chiu et al. [14] showed in their meta-analysis and systematic review a non-linear association between sleep duration and the odds to report symptoms of suicidality: with increasing sleep duration from less than 6 h to about 8.5 h, the odds to report symptoms of suicidality decreased, while the same odds increased with increasing sleep durations from 8.5 h to 12 h. This pattern of results implied two conclusions: first, sleep duration per se should be used cautiously, and this holds particularly true when sleep duration exceeds the normative amount of about 9 h/night for adolescents [15]. Second, there is increasing evidence of non-linear associations between sleep duration and psychological functioning [16,17,18,19]. Given this, assessing sleep quality appears to be associated with cognitive–emotional processes in a tighter relation, compared to sleep duration. This observation has already been reported elsewhere [20].

As regards the research on sleep among adolescents in Iran, publications are less abundant: Khazaie et al. [5] assessed a sample of 300 about 16-years old adolescents; the results showed that risky behavior and susceptibility to substance use were associated with poor attention to sleep hygiene rules, procrastinating sleep onset, or excessively psychologically arousing activities before bedtime. Farhangi et al. [21] showed among a sample of 80 12- to 16-years old adolescents that both higher night eating and emotional eating were associated with lower sleep quality. To explore cognitive–emotional and behavioral processes underlying sleep hygiene behavior among a sample of 1822 healthy adolescents (mean age = 13.97 years) Strong et al. [22] applied the Theory of Planned Behavior [23,24] and showed that knowledge about sleep hygiene did *not* improve intention and behavior to use and improve favorable sleep behavior. Strong et al. [22] concluded that teaching sleep hygiene appeared to be insufficient to improve Iranian adolescents’ favorable sleep behavior, while coping with poor sleep and action planning to improve sleep should be more successful.

Further, to assess subjective sleep among Iranian adolescents, often either the Insomnia Severity Index [25,26,27,28,29,30], the Adolescent Sleep Hygiene Scale [5,31,32], or the Pittsburgh Sleep Hygiene Index for adults [5,32,33] are employed.

The Persian versions of the Adolescent Sleep Hygiene Scale [31,32], the Insomnia Severity Index for Adults [34], and the Adult Pittsburgh Sleep Quality Index [35,36] have been previously validated. In the present study, we established the psychometric properties of the Persian Pittsburgh Sleep Quality Index (PSQI) for adolescents. The reasons are as follows: 1. The PSQI is one of the internationally most employed self-rating tools to assess sleep quality; 2. The English [37] and Portuguese [38] version of the PSQI for adolescents have been validated; 3. The Persian version of the PSQI has already been employed among adolescents; 4. Its psychometric validation appears to be overdue.

Given this, as the original PSQI [39], the Persian PSQI for adolescents is aimed at offering a quick and reliable tool to assess self-rated sleep quality.

## 2. Methods

### 2.1. Study Procedure

Young adolescents attending the high schools of Kermanshah (Iran) were approached to participate in the present cross-sectional study. Eligible participants were fully informed about the aims of the study and the anonymous data handling. Thereafter, they signed the written informed consent. For participants younger than 14 years, signed written informed consent was asked from their legal guardians. Participants completed a series of paper-and-pencil questionnaires covering sociodemographic and sleep-related information. The local ethics committee of the Kermanshah University of Medical Sciences (KUMS, Kermanshah, Iran) approved the study (code: ir.kums.rec. 1397.676), which was performed in accordance with the ethical principles laid down in the seventh and current edition [40] of the Declaration of Helsinki.

### 2.2. Sample

A total of 1500 adolescents were approached. Of those, 1440 participants (96%) agreed to participate (46.8% males), while 60 (4%) did not complete the written informed consent or did not answer all questions. The mean age was 15.74 years (SD = 1.82; range: 11–18 years). A total of 628 (42.6%) were attending the first level of secondary school, and 849 (57%) were attending the second level of secondary school. Two hundred (13.5%) reported smoking tobacco or hookah on occasion during leisure time. The majority of participants (n = 1242; 84.1%) were living with their parents; 145 (9.8%) were living with their mother; 32 (2.2%) were living with their father; 58 (3.9%) did not report on the family status.

### 2.3. Tools

#### 2.3.1. Sociodemographic Information

Participants reported their age (years), sex (male/female), vocational status, family status, and use of tobacco/hookah (see below).

#### 2.3.2. Pittsburgh Sleep Quality Index

The English original questionnaire [39] was translated into Persian. To do so, we followed the algorithm proposed by Brislin [41] and Beaton et al. [42]. First, two English and Persian speaking translators translated independently the items. Second, the two versions of translated items were compared. In the case of full linguistic and semantic overlap, the item remained unchanged. In the case of low linguistic or semantic overlap, a third translator was introduced to find the best linguistic or semantic fit between divergent translations. Third, two independent translators back-translated the Persian version into English. Fourth, in case of high linguistic and semantic overlap between the original English items and the translated and back-translated version, Persian items were accepted as the final version. In the case of linguistic or semantic differences, both the Persian and the translated English versions were adapted, till high linguistic and semantic overlap was found.

Briefly, the PSQI is an 18-item self-report questionnaire designed to assess overall sleep quality over a one-month period [39]. As very well outlined elsewhere [37], the first four items relate to sleeping habits and are in a free-response format (e.g., “How many hours of actual sleep do you get at night?”). The remaining items are related to sleep disturbances and daytime impairments. These items are rated in terms of the frequency or severity of the problem. Answers are given on four-point Likert scales with the following anchor points: 0 (= not during the past month), 1 (= less than once a week); 2 (= once or twice a week); 3 (= three or more times a week). Further, we followed Raniti et al. [37] and modified the wording of item 8 as follows: While the original wording is “How often did you have trouble staying awake *while driving*, eating meals, or engaging in social activity.”, the current wording was “How often did you have trouble staying awake *during class*, eating meals, or engaging in social activity.”. The 18 items are summed up to the following dimensions: 1. Subjective Sleep Quality; 2. Sleep Latency; 3. Sleep Duration; 4. Habitual Sleep Efficiency; 5. Sleep Disturbances; 6. Sleeping Medication; 7. Daytime Dysfunction. Each component score has a possible range of 0 to 3, with higher scores indicating poorer sleep. Further, the PSQI global score of sleep quality ranges from 0 to 21. A sum score of 5 points or higher reflects sleep disturbances (Cronbach’s alpha = 0.81).

#### 2.3.3. Adolescent Sleep Hygiene Scale

Participants completed the revised version of the Adolescent Sleep Hygiene Scale [31,43,44]. The questionnaire consists of 24 items, loading on the following six categories: 1. Physiological factor (5 items; e.g., “I go to bed feeling hungry”; “After 6:00 p.m., I have drinks with caffeine (e.g., cola, iced tea, coffee”); 2. Behavior Arousal factor (3 items; e.g., “I go to bed and do things in my bed that keep me awake (e.g., watching TV, reading)”; “I use my bed for things other than sleep (e.g., talking on the telephone, watching TV, playing video games, doing homework)”; 3. Cognitive–emotional factor (6 items: e.g., “I go to bed and think about things I need to do.”; “I go to bed and worry about things at home or at school.”); 4. Daytime sleep factor (2 items; “During the day, I take a nap that lasts > 1 h.”; “After 6:00 p.m., I take a nap.”); 5. Sleep environment factor (5 items; e.g., “I fall asleep in a room that feels too hot or too cold.”; “I fall asleep in one place and then move to another place during the night.”); 6. Sleep stability factor (3 items; e.g., “On weekends, I sleep in more than 1 h past my usual wake time.”). Answers are given on a six-point Likert scale ranging from 1 (= never) to 6 (always), and with higher mean scores reflecting a more pronounced dimension (overall Cronbach’s alpha = 0.82).

### 2.4. Statistical Analysis

First, item clusters were calculated, following the manual of the PSQI [39].

Second, a series of Pearson’s correlations was performed to calculate the strengths of association between the item clusters.

Third, with a series of Pearson’s correlations associations between the item clusters of the PSQI and the clusters of the Adolescent Sleep Hygiene Scale (ASHS) were calculated.

Fourth, with a series of t-tests differences of ASHS scores were calculated between participants with normal sleep (PSQI ≤ 5 points) and with impaired sleep (PSQI > 6 points).

Fifth, item clusters of the PSQI were further reduced in their dimensions. To do so, a factor analysis was performed.

Sixth, the newly defined factors were employed to predict the overall score of the ASHS.

Seventh and last, a first-order Confirmatory Factor Analysis (CFA) was carried out to test the factorial validity of the PSQI. We used maximum likelihood (ML) for parameter estimation, and considered multiple goodness-of-fit indices to find out whether the theoretical model fitted well with the empirical data [45]. Based on the recommendations of Byrne [46], model fit was considered adequate if the Normed Fit Index (NFI) was ≥0.90, Comparative Fit Index (CFI) was ≥0.90, Tucker–Lewis Index (TLI) was ≥0.90, and if the RMSEA was ≤0.05. Following Comrey and Lee [47], standardized factor loadings of ≥0.71 were considered as excellent, ≥0.63 as very good, ≥0.55 as good, and ≥0.45 as fair. Cronbach’s alphas were then calculated to test internal consistency. Significant correlations of r < 0.30 are interpreted as small, of r = 0.30 to 0.50 as medium, and of r > 0.50 as large [48]. CFA was carried out with AMOS^®^ 22 (IBM Corporation, Armonk, NY, USA); all other analyses were performed with SPSS^®^ 25.0 (IBM Corporation, Armonk, NY, USA) for Apple Mac^®^.

## 3. Results

### 3.1. Overview of the Dimensions of the Pittsburgh Sleep Quality Index (PSQI)

Classically, the PSQI yields eight dimensions: Subjective Sleep Quality, Sleep Onset Latency, Sleep duration, Habitual Sleep Efficiency, Sleep Disturbances, Use of sleeping medication; Daytime Dysfunction, along with an overall score; higher scores always reflect a more pronounced level: e.g., higher scores of subjective sleep quality mirror a *more impaired* sleep quality. However, to easily grasp the content and the directions of associations, the wording was adapted. Table 1 shows the descriptive statistical indices, along with the correlation coefficients. Statistical indices are not repeated in the text again. Further, we followed Raniti et al. [37] and report in Table 2 (see below) the Cronbach’s alphas for the PSQI scores (“if item is deleted”).

A lower Subjective Sleep Quality (that is to say: a higher score of Subjective sleep quality) was associated with longer Sleep Onset Latency, shorter Sleep Duration, lower Habitual Sleep Efficiency, higher Sleep Disturbances, a more frequent Use of Sleep Medications, a more impaired Daytime Functioning, and an overall lower Sleep Quality.

A longer Sleep Onset Latency was associated with a shorter Sleep Duration, a lower Habitual Sleep Efficiency, more frequent Sleep Disturbances, a more frequent Use of Sleep Medications, a higher Daytime Dysfunction, and an overall lower Sleep Quality.

A longer Sleep Duration was associated with a lower Habitual Sleep Efficiency, higher Sleep Disturbances, a more frequent Use of Sleep Medications, a higher Daytime Dysfunction, and an overall lower Sleep Quality.

A lower Habitual Sleep Efficiency was associated with higher Sleep Disturbances, a more frequent Use of Sleep Medications, a higher Daytime Dysfunction, and an overall lower Sleep Quality.

Higher Sleep Disturbances were associated with a more frequent Use of Sleep Medications, a higher Daytime Dysfunction, and an overall lower Sleep Quality.

A higher Use of Sleep Medications was associated with a more impaired Daytime Functioning, and an overall lower Sleep Quality.

A higher Daytime Dysfunction was associated with an overall lower Sleep Quality.

### 3.2. Correlations between Dimensions of the PSQI with Dimensions from the Adolescent Sleep Hygiene Scale (ASHS)

Table 3 reports the correlation coefficients between dimensions of the PSQI and the Adolescent Sleep Hygiene Scale (ASHS). Statistical indices are not repeated in the text again.

A lower Subjective Sleep Quality (i.e., a higher score on Subjective Sleep Quality) was associated with higher scores of Physiological, Behavioral–Arousing, Cognitive–emotional factors of poor sleep hygiene, along with higher Daytime Sleepiness, poor Sleep Environment and poor Sleep stability.

A longer Sleep Onset Latency was associated with higher Cognitive–emotional factors of poor sleep, along with poor Sleep stability. No meaningful associations (that is: correlation coefficients of 0.125 and lower) were found for Physiological, Behavioral–Arousing and Sleep Environment-related factors.

A shorter Sleep Duration was associated with higher Physiological, Behavioral–Arousing, Cognitive–emotional factors of poor sleep hygiene, along with a higher Daytime Sleepiness. No meaningful associations (that is: correlation coefficients of 0.125 and lower) were found for Physiological, and Sleep Environment-related and Sleep Stability-related factors.

A lower Habitual Sleep Efficiency was not meaningfully associated with any dimension of the ASHS.

Higher Sleep Disturbances were associated with higher Physiological, Behavioral–Arousing, Cognitive–emotional factors of poor sleep hygiene, along with higher Daytime Sleepiness, poor Sleep Environment and poor Sleep stability.

A higher Use of Sleep Medication was associated with higher Physiological, Behavioral–Arousing, Cognitive–emotional factors of poor sleep hygiene, along with higher Daytime Sleepiness, poor Sleep Environment and poor Sleep stability.

A higher Daytime Dysfunction was associated with higher Physiological, Behavioral–Arousing, Cognitive–emotional factors of poor sleep hygiene, along with higher Daytime Sleepiness, poor Sleep Environment and poor Sleep stability.

A lower overall Sleep Quality was associated with Physiological, Behavioral–Arousing, Cognitive–emotional factors of poor sleep hygiene, along with higher Daytime Sleepiness, poor Sleep Environment and poor Sleep stability.

To summarize, dimensions of the PSQI and dimension of the ASHS correlated and showed an overlap, though correlation coefficients were small to medium.

### 3.3. Dimensions of the Adolescent Sleep Hygiene Scale in Participants with Normal (PSQI ≤ 5) and with Poor Sleep (PSQI > 6)

Table 4 reports the descriptive and statistical indices of ASHS scores between normal and poor sleepers.

Compared to normal sleepers (n = 729; 49.4%), poor sleepers (n = 748; 50.6%) reported statistically significantly higher scores of Physiological, Behavioral–Arousing, Cognitive–Emotional factors of poor sleep hygiene, along with higher Daytime Sleepiness, poor Sleep Environment and poor Sleep stability. However, effect sizes were small to medium.

### 3.4. Factor Analysis of Dimensions of PSQI

To further merge the dimensions of the PSQI, a factor analysis was performed. Standard criteria for such an analysis were met: (a) the minimum sample size of 100 participants was exceeded; (b) all items were scored in the same direction; (c) correlations were high and significant (see also Table 1); (d) the Kaiser–Meyer–Olkin (KMO) Measure of Sampling Adequacy was acceptable at 0.72 > 0.05; (e) Bartlett’s Test of Sphericity was significant (X^2^(N = 1440; df = 21) = 1088.03, *p* < 0.0001). Factors were extracted with Principal Component Analysis (PCA). Factors were extracted with Eigenvalues greater than 0.75, and orthogonal varimax rotation was employed.

The exploratory factor analysis yielded four factors with Eigenvalues > 0.75, together explaining 72.72% of the total variance. High Sleep Disturbances, Medication Use and Low Daytime Function (31.76% of total variance); short Sleep Duration and low Sleep Efficiency (16.1% of total variance); Short Sleep Duration and poor Sleep Quality (13.46% of total variance); long Sleep Onset Latency and Low Daytime Function (11.23% of total variance). To conclude, the dimensions of the PSQI could be further merged into four distinguishable factors (see also Table 5).

### 3.5. Multiple Regression Analysis to Predict Scores of the Adolescent Sleep Hygiene Scale, Based on the Four Factors of the PSQI

The four factors derived from the factor analysis (see above) were employed to predict the overall score of the ASHS.

First, preliminary statistical requirements to run a multiple regression analysis were met: 1. The sample size was sufficiently high; 2. The Durbin–Watson coefficient indicated that residuals of the predictors were independent (1.57; range should be between 1.5 and 2.5); 3. Predictors sufficiently explained the variance of the dependent variable (R = 0.485; R^2^ = 0.235).

As shown in Table 6, higher scores on High Sleep Disturbances, Medication Use and Low Daytime Function, Short Sleep Duration and poor Sleep Quality, and long Sleep Onset Latency and Low Daytime Function independently predicted higher scores of ASHS, while short Sleep Duration and low Sleep Efficiency was excluded from the equation, as this factor did not reach statistical significance. 

### 3.6. Model Fit of the PSQI: Results from the Structural Equation Modelling

Overall, the model fit was satisfactory, with the following statistical indices: Normed Fit Index (NFI; should be ≥0.90): 0.901; Composite Fit Index (CFI; should be ≥0.90): 0.918; Tucker–Lewis Index (TLI; should be ≥0.90): 0.875; Root Mean Square of Approximation (RMSEA; should be ≤0.05): 0.043 (see also Figure 1).

## 4. Discussion

To the best of our knowledge, this is the first study to examine the psychometric properties of the Persian language version of the PSQI in a community sample of adolescents. The key findings of the present study were that among a larger sample of adolescents the Persian version of the Pittsburgh Sleep Quality Index (PSQI) for adolescents had acceptable and satisfactory psychometric properties, that is to say, the first-order CFA to test the factorial validity of the PSQI showed satisfactory indices. Further, when compared to the well-established Adolescent Sleep Hygiene Scale, significant correlations were found, supporting thus the concurrent validity of the PSQI, even if the magnitude of the correlation coefficients were small to medium. Next, the cut-off-point of five points and higher indicated that 50.6% of participants complained about poor sleep, which again was mirrored in more impaired sleep values, as assessed with the ASHS. Last, to further merge the factorial structure of the PSQI, a four-factorial solution was found, which predicted the overall score of poor sleep, as assessed with the ASHS.

The CFA confirmed the original seven-factor solution (see Figure 1), with acceptable and satisfactory model-fit indices. It follows that the Persian version appears to be suitable not only for adults [49], but also for adolescents. Accordingly, the PSQI might be employed in future studies as an easy-to-complete and internationally well accepted self-rating questionnaire to assess subjective sleep quality.

To calculate the concurrent validity of the PSQI, the ASHS was employed. As shown in Table 2, correlation coefficients were statistically significant, but were generally of small to medium magnitude. The quality of the data does not allow a deeper understanding as to why correlation coefficients were small to medium. One reason, however, might be that the ASHS is much more focused on cognitive–emotional and behavioral factors to explain sleep quality, while the PSQI is more focused on the structure of sleep patterns. It follows that using the ASHS as a concurrent questionnaire might not have been the best choice to validate the PSQI. However, to our knowledge, there was no alternative questionnaire available. As regards the Insomnia Severity Index (ISI, [25]), to our knowledge, the Persian version has been validated for adults [34,50], but not for adolescents so far. Further, strictly taken, symptoms of insomnia are not equal to sleep complaints as a whole.

The seven-factor structure found in the CFA is not in line with previous studies carried out with Australian [37] or Brazilian [38] adolescents; Passos et al. [38] calculated a two-factor model to reflect most appropriately Brazilian adolescents’ sleep patterns, while Raniti et al. [37] calculated a one-factor solution for Australian adolescents.

To further explore the factor structure of the present data, a further factor analysis yielded a four-factor solution, although factors with Eigenvalues above 0.75 were also accepted. As shown in Table 4, factorial coefficients were between 0.56 and 0.70, indicating robust patterns of association.

Despite the encouraging pattern of results, the following limitations should be considered. First, the concurrent and cross-validation showed small to medium magnitudes of associations. It follows that instead of the ASHS, a questionnaire closer to the pattern of the PSQI might have yielded more robust results to assess the concurrent validity of the PSQI. However, as mentioned before, we are unaware of such a self-rating questionnaire for adolescents to assess their sleep quality. Second and similarly, it would have been important and interesting to cross-validate sleep patterns with further psychological dimensions such as symptoms of depression, anxiety and daytime sleepiness; results from previous studies showed that poor sleep is associated with higher symptoms of depression, anxiety [51,52,53] and daytime sleepiness [54,55,56]. Third, participants were not screened for psychiatric disorders, which might have biased the entire pattern of results. To illustrate, when we consider a prevalence rate of 5.6% of children and adolescents with attention-deficit/hyperactivity disorder (ADHD) [57,58], mathematically, 81 participants out of 1440 should be diagnosed with ADHD; or on other words: It is highly conceivable that within a sample of 1440, subgroups with different patterns of results might have been detectable. It follows that the present patterns of results might be biased due to latent and unassessed psychological, psychiatric or neurophysiological dimensions.

## 5. Conclusions

The pattern of results suggests that the Persian version of the PSQI for adolescents has satisfactory psychometric properties to assess adolescents’ sleep quality. Given this, it appears that the Persian PSQI for adolescents is a useful, easy-to-complete self-rating questionnaire for sleep quality. Further, the present questionnaire sets the ground for future studies in the field of adolescent sleep quality not only in Iran, but also for the scientific community interested in research on adolescent sleep.

## Figures and Tables

**Figure 1 ijerph-17-07095-f001:**
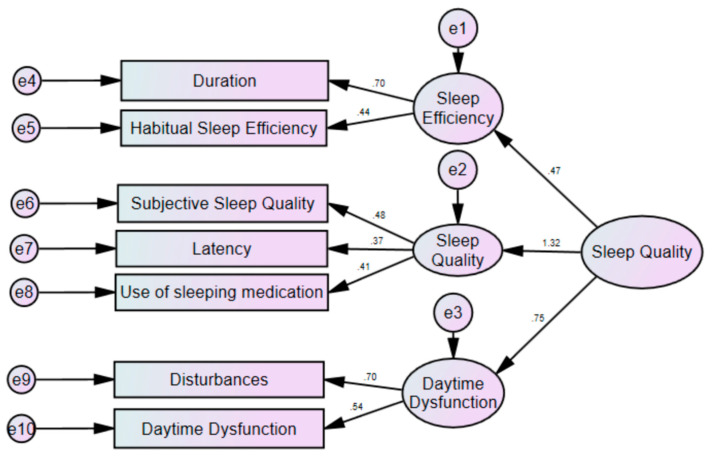
Model fit of the Pittsburgh Sleep Quality Index (PSQI); results from the structural equation modelling.

**Table 1 ijerph-17-07095-t001:** Overview of the intercorrelations between the seven dimensions of the Pittsburgh Sleep Quality Index (PSQI).

			Dimensions					
	Subjective Sleep Quality	Sleep Onset Latency	Sleep Duration	Habitual Sleep Efficiency	Sleep Disturbances	Medication Use	Daytime Dysfunction	Overall Sleep Quality
Subjective sleep quality	-	0.24 ***	0.26 ***	0.11 **	0.31 ***	0.19 ***	0.23 ***	0.59 ***
Sleep onset latency		-	0.16 ***	0.07 *	0.25 ***	0.09 *	0.24 ***	0.52 ***
Sleep duration			-	0.31 ***	0.20 **	0.11 *	0.11 *	0.55 ***
Habitual sleep efficiency				-	0.09 *	0.17 **	0.10 *	0.52 ***
Sleep disturbances					-	0.32 ***	0.38 ***	0.60 ***
Medication use						-	0.23 **	0.54 ***
Daytime dysfunction							-	0.58 ***
Overall sleep quality								-

Notes: * = *p* < 0.05; ** = *p* < 0.01; *** = *p* < 0.001.

**Table 2 ijerph-17-07095-t002:** Cronbach’s alphas for the PSQI components.

	Component	Cronbach’s Alpha When Item Deleted
1	Subjective sleep quality	0.77
2	Sleep latency	0.71
3	Sleep duration	0.69
4	Sleep efficiency	0.79
5	Sleep disturbance	0.79
6	Use of sleep medication	0.77
7	Daytime dysfunction	0.070
	Cronbach’s alpha	0.73

**Table 3 ijerph-17-07095-t003:** Overview of the intercorrelations between the seven dimensions of the Pittsburgh Sleep Quality Index (PSQI); N = 1477.

			Adolescent Sleep Hygiene Scale Factors			
	Physiological	Behavioral–Arousing	Cognitive–Emotional	Daytime Sleepiness	Sleep Environment	Sleep Stability
Pittsburgh Sleep Quality Index						
Subjective sleep quality	0.15 **	0.25 ***	0.26 ***	0.27 ***	0.16 ***	0.21 ***
Sleep onset latency	0.00	0.07 *	0.21 ***	0.13 **	0.01	0.15 ***
Sleep duration	0.11 **	0.15 ***	0.19 ***	0.15 **	0.05	0.10 *
Habitual sleep efficiency	0.02	0.06	0.06	0.06	0.06	0.09 *
Sleep disturbances	0.24 ***	0.30 ***	0.33 ***	0.35 ***	0.29 ***	0.23 ***
Medication use	0.30 ***	0.31 ***	0.23 ***	0.25 ***	0.30 ***	0.17 ***
Daytime dysfunction	0.15 *	0.17 **	0.24 ***	0.21 ***	0.19 ***	0.18 ***
Overall sleep quality	0.24 ***	0.33 ***	0.38 ***	0.35 ***	0.26 **	0.28 ***

Notes: * = *p* < 0.05; ** = *p* < 0.01; *** = *p* < 0.001.

**Table 4 ijerph-17-07095-t004:** Descriptive and inferential statistical indices of the dimensions of the Adolescent Sleep Hygiene Scale (ASHS), separately for good sleepers (n = 729) and poor sleepers (n = 748).

	Pittsburgh Sleep Quality Index; Categories			
	Good Sleepers	Poor Sleepers		
N	729	748		
	M (SD)	M (SD)	*t*-tests	Cohen’s d
Dimensions of Adolescent Sleep Hygiene Scale				
Physiological	12.93 (4.27)	14.66 (4.96)	t(1475) = 7.15 ***	0.37 (S)
Behavioral–arousing	6.22 (2.78)	7.80 (3.35)	t(1475) = 9.83 ***	0.51 (M)
Cognitive–emotional	18.74 (5.81)	22.12 (5.79	t(1475) = 11.21 ***	0.58 (M)
Daytime sleepiness	5.29 (2.43)	6.92 (2.83)	t(1475) = 11.86 ***	0.61 (M)
Sleep environment	11.18 (4.69)	13.58 (5.43)	t(1475) = 9.09 ***	0.47 (S)
Sleep stability	9.09 (3.38)	10.68 (3.49)	t(1475) = 8.91 ***	0.46 (S)

Notes: S = small effect size; M = medium effect size. *** = *p* < 0.001.

**Table 5 ijerph-17-07095-t005:** Factor analysis: Four-factor solution of the seven dimensions of the Pittsburgh Sleep Quality Index (PSQI); correlation coefficients.

		New Factors			
Original Dimensions		High Sleep Disturbances, Medication Use and Low Daytime Function	Short Sleep Duration and Low Sleep Efficiency	Short Sleep Duration and Poor Sleep Quality	Long Sleep Onset Latency and Low Daytime Function
Overall variance	72.72%	31.8%	16.1%	13.5%	11.2%
Subjective sleep quality				0.833	
Sleep onset latency					0.867
Sleep Duration			0.581	0.603	
Habitual sleep efficiency			0.906		
Sleep disturbances		0.673			
Use of sleep medications		0.807			
Daytime dysfunction		0.613			0.505

Note: The overall score termed Sleep Quality was not entered into the factor analysis.

**Table 6 ijerph-17-07095-t006:** Multiple linear regression with the new factors of the Pittsburgh Sleep Quality Index as predictors and the Adolescent Sleep Hygiene Scale (overall score) as dependent variable.

Dimension	Variables	Coefficient	Standard Error	Coefficient β	t	*p*	R	R^2^	Durbin–Watson Coefficient
ASHS total score	Intercept	11.61	0.072	-	160.851	0.000	0.485	0.235	1.572
	High sleep disturbances, medication use and low daytime function	1.345	0.072	0.424	18.623	0.000			
	Short sleep duration and low sleep efficiency	0.720	0.072	0.227	9.967	0.000			
	Long sleep onset latency and low daytime function	0.189	0.072	0.060	2.616	0.006			
Excluded variable:	Short sleep duration and poor sleep quality: *p* > 0.05								

Notes: ASHS = Adolescent Sleep Hygiene Scale.

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
