# Peer review of "Psychometric Properties of the Persian Pittsburgh Sleep Quality Index for Adolescents"

_ijerph, 2020, doi:10.3390/ijerph17197095_

Round 1
Reviewer 1 Report
The authors have to a reasonable degree adressed the concerns raised by the reviewers. I still believe the manuscript would benefit from cutting the introduction from line 36 to 49, and maintain the focus on validating the instrument from the start of the introduction. Minor rewrites from 50 and onwards may be necessary as well.
I would also question the statement of aims as "exploring the psychometric properties". It seems to me that they are trying to establish that the psychometric properties of the translated version are compareable to the original instrument, which is a reasonable and perfectly valid aim, and their introduction and discussion should reflect this.
Author Response
Thank you for your valuable comments.
Please find attached the detailed point-by-point-response and the corrected manuscript.

Reviewer 2 Report
The authors have followed the recommendations and suggestions of the reviewers and have made appropriate changes along the different section of the manuscript that have improved the quality of the work presented.
However, I encourage the authors to check the bibliographical references as there are quite a few errors.
Author Response

(The authors gave the same response as above.)

Reviewer 3 Report
I am satisfied with the authors response to comments.
Author Response

(The authors gave the same response as above.)

Reviewer 4 Report
I appreciate the opportunity to review this study. The theme becomes even more important given the impact of the situation of social isolation on the human behavior. Thus, the article seems very timely, adjusting to the new issues related to the sleep quality of adolescents.
The structure of the article is adequate, it is clear in the presentation of the objective and the method is adequate to achieve that objective.
I could see that there have already been some changes in the text (in bold) and I think that all of them were very well done, further improving the quality of the manuscript.
A single doubt I have is regarding the repeated use of the term Farsi / Persian to designate the language into which the questionnaire was translated. I suggest that at some point in the text the authors explain why they chose to use these denominations together.
Regarding the Results, Figure 1 needs clear editing, as there are cuts in the arrows and there is no title for this figure.
In the Discussion section, it can be seen that the authors show mastery of the theme, and the writing is very simple and elegant, as a scientific text should be.
The conclusion is adequate, consistent with what was the objective of the study.
Author Response
Thank you for your valuable comments.
Please find attached the detailed point-by-point-response and the corrected manuscript.

This manuscript is a resubmission of an earlier submission. The following is a list of the peer review reports and author responses from that submission.
Round 1
Reviewer 1 Report
The paper presents a psychometric study of the Farsi version of the Pittsburgh Sleep Quality Index for adolescents in an Iranian sample. I would commend the authors for conducting the psychometric groundwork to enable the use of this instrument in Farsi, and for collecting a solid sample size. The methods section and the results are quite clearly presented, and the analysis seems to be adequately conducted. I still have some major concerns about the paper as it stands.
The introduction should more clearly state what the authors mean by validating the measure. The measurement construct should be clarified (what is sleep quality, and how is it a different construct from insomnia, sleep habits and sleep hygiene?) and the authors should state what uses of the instrument they aim to validate. Is it the use of a total score (or the use of specific factor scores) as a screener for impaired sleep, as a continuous measure of sleep impairment, or as a measure of change in sleep impairment? These uses have different requirements for psychometric properties and would require different psychometric analyses. The analyses conducted and reported then need to support the stated aims. For the reader, it is important to know what uses of the instrument has gotten support from this study.
The authors should carefully reconsider the utility of the ASHS as a measure to establish concurrent validity; they already discuss this, but I am not convinced by their argument that it is suitable because other measures are not available. It is also unclear whether the construct validity of the PSQI is really in question after a translation to Farsi?
I also encourage the authors to look to the way Raniti and colleagues (in a paper from 2018 which they cite) used cross-validation, as they have a sample size sufficient for doing so as well.
Further, some of the authors have published other papers on sleep in adolescent samples, and the methods section (2.2) should specify explicitly whether this sample is entirely new, or cite any previous studies conducted on the same dataset, or parts of this dataset.
Reviewer 2 Report
The paper is interesting. I can see the regional relevance for translating and validating this research instrument. However, there are some points that need urgent clarification:
Ethical approval - It would seem from Section 2.1 that the study was approved by a research ethics committee, however the process for achieving consent was definitely not in line with the Declaration of Helsinki. Authors state that "Young adolescents attending the high schools of Kermanshah (Iran) were approached to participate in the present cross-sectional study. Eligible participants were fully informed about the aims of the study and the anonymous data handling. Thereafter, they signed the written informed" - so there was no involvement of parents/legal guardians, which was mandatory. DoH indeed states that "Whenever the minor child is in fact able to give a consent, the minor’s consent must be obtained in addition to the consent of the minor’s legal guardian.” To my understanding this study is not ethically cleared.
The statistical approach is not robust - it relies on a parametric test (Pearson's correlation), but the variables are categorical Likert scales (not ideal) and we don't know about distributional patterns of the variables scores so we cannot tell if assumptions for parametric test were met or violated. Non-parametric approach should have been sought. Thus, I cannot assess the validity of the study.
Finally, if the study is to validate a Farsi translation of a research instrument, I would have expected (at least as an appendix) the actual protocol in Farsi to be reported.
Reviewer 3 Report
Many thanks for allowing me to review this interesting manuscript.
This manuscript explores the psychological properties of Farsi Pittsburgh Sleep Quality Index for Adolescents
In general, the article is very well developed, the introduction is interesting, but need to make a better selection of bibliographic references and the methodology and procedure are adequate to analyze the properties of the scale. The discussion is in accordance with the object of study.
Despite the interest of the work, it presents some major aspects, that the correction of them will improve the quality of work presented.
Changes or suggestions relate to:
Introduction and abstract
Although the introduction is adequate, it needs reduce the number of references and make a better selection of them. The references must be more specific and less general.
Authors need to better describe the objectives of the work, both in the introduction and in the abstract
Material and methods
Study design
Some procedural aspects do not go in this design section but in procedure section, please create a procedure subsection.
Sample and procedure
How were the participants selected? authors should better describe this aspect. How were the questionnaires administered? The survey was online or pencil and paper. How was the sample calculated? How many potential subjects and how many refused to participate, or how many responded and how many responses were incomplete?
In the second paragraph of this section, after describing the sample, the authors present results of comparing gender with other variables. This section should go to results.
Results
Given the characteristics of the study the analyses were adequate.
One of the important aspects is the reliability of the scale, the authors must present a table with these data (item-total correlations, alpha if item is deleted, alpha of the different dimensions of the scale, etc.
In lines 237-8 there is a mistake in the criteria to assign people to different groups levels. Please revise this aspect.
In Table 4 please delete the note of levels of significance due that the authors have presented p values in the table.
Discussion
New selected references are needed in discussion to support the findings of the study.
It is necessary to describe in greater detail the practical applications of the finding and the supposed benefit that it can bring to adolescent population to improve their quality of sleep
References
The references seem appropriate but need an update.
Please revise some references that doesn't meet the Vancouver criteria.
Reviewer 4 Report
The manuscript describes a validation of the Farsi/Persian Pittsburgh Sleep Quality Index.
The PSQI is a commonly used scale for assessing sleep quality, it makes sense to validate and make this measure available to other languages and cultures, though there are already studies that have deployed it as mentioned in the article.
On lines 69-70, dysfunctional is used three times in one sentence. Revise for redundancy.
On line 121 there is a sentence fragment "Briefly, the PSQL consists of"
The procedures for translation make sense except for the reasoning for item 8 (line 128-131). Please make clear the significance of this change and why it was done.
On line 139 the authors state 24 items "loaded" onto six factors. This implies an analysis, but I think they meant to say six categories instead?
Correlations don't seem very high for the PQSI intercorrelations in Table 1 or for the PQSI-ASHS in table 2. Is this surprising? It's mentioned in the abstract, but wouldn't it make sense to include data from English versions to more easily compare and contrast? Has noone else empirically examined the arguement for weak correlations between ASHS and PSQI?
Conclusions section seems inadequate and short.